# Research on Calendar Aging for Lithium-Ion Batteries Used in Uninterruptible Power Supply System Based on Particle Filtering

**Wei Xu ***  **and Hongzhi Tan**

Shanghai Electric Group Co., Ltd., Central Academy, Shanghai 200070, China
* Correspondence: xuwei_ap@126.com

**Abstract:** The aging process of lithium-ion batteries is an extremely complex process, and the prediction of the calendar life of the lithium-ion battery is important to further guide battery maintenance, extend the battery life and reduce the risk of battery use. In the uninterruptible power supply (UPS) system, the battery is in a floating state for a long time, so the aging of the battery is approximated by calendar aging, and its decay rate is slow and difficult to estimate accurately. This paper proposes a particle filtering-based algorithm for battery state-of-health (SOH) and remaining useful life (RUL) predictions. First, the calendar aging modeling for the batteries used in the UPS system for the Shanghai rail transportation energy storage power station is presented. Then, the particle filtering algorithm is employed for the SOH estimation and RUL prediction for the single-cell battery calendar aging model. Finally, the single-cell SOH and RUL estimation algorithm is expanded to the pack and group scales estimation. The experimental results indicate that the proposed method can achieve accurate SOH estimation and RUL prediction results.

**Keywords:** lithium-ion battery; battery health prediction; calendar aging; particle filtering

## 1. Introduction

### 1.1. Research Background and Motivation

The improvements in lithium-ion battery technology have achieved a longer range and life. The optimization of lithium-ion batteries in the traditional energy structure, which is dominated by fossil energy sources such as oil and coal, has promoted the flourishing development of new energy vehicles, smart grids and other environment-friendly industries [1]. However, taking into account the effects of battery manufacturing, operating conditions and environmental conditions, lithium-ion batteries inevitably experience continuous deterioration in performance during use, even causing uncontrolled combustions or explosions. Therefore, with the goal of safety and efficiency, it is of great value to study the model for the internal health states of lithium-ion batteries and accurate state estimation methods [2].

The aging process of lithium-ion batteries is an extremely complex process, and the prediction of its life requires not only empirical values and data accumulation about the battery but also a model based on the aging mechanism of the battery to predict more accurately [3,4]. The calendar life prediction of the lithium-ion battery is important to further guide the battery maintenance, extend the battery life and reduce the risk of battery use. The calendar aging for lithium-ion batteries used in the uninterruptible power supply (UPS) system is hard to estimate because of the slow decay rate of the battery, and it is difficult to find measurable decay characteristics. This paper proposes a particle filtering-based algorithm for battery state-of-health (SOH) and remaining useful life (RUL) estimation.

*1.2. Literature Review*

One of the important indicators of battery performance is the degree of battery aging, which is mainly affected by the charging and discharging cycle process impact. In this type of process, battery components such as electrodes will be affected by the aging process, thus affecting working conditions and safety. Battery capacity degradation is a widely accepted indicator for battery aging evaluation and is also an important indicator for the operation of systems using batteries as energy sources. Once the battery capacity decays to a certain threshold, the battery can be considered to have reached the upper limit of its life, and its safety and performance are insufficient to maintain its continued use. Therefore, it must be replaced. In the UPS system, the battery is in a floating state for a long time, so the aging of the battery is approximated by calendar aging, and its decay rate is slow and difficult to estimate accurately.

The prediction methods for SOH can usually be divided into direct measurement methods, model-based methods and data-driven methods.

The direct measurement method is widely used for the simple testing of the SOH of lithium-ion batteries [5]. For the Coulomb counting method, the accuracy depends on the accurate measurement of the cell current and the accurate estimation of the initial SOC. A fully charged battery has a maximum releasable capacity, and the difference from the rated capacity can be used to assess the capacity degradation of the battery. The internal-resistance-based estimation methods use internal resistance to map the capacity of lithium-ion batteries and thus estimate the SOH. The internal resistance of a battery can be measured by mixing pulse power characteristics, which have a low computational complexity but are less robust as an open-loop measurement method and are more sensitive to measurement noise [6].

In order to overcome the drawbacks of the direct measurement method, scholars have proposed methods to describe the relationship between the internal resistance and the available capacity based on the equivalent circuit model. For lithium-ion batteries, the equivalent circuit model has some parameters to be determined, and the parameters of the model can be identified based on the relevant testing data of the battery [7,8]. There are also some more typical methods such as battery SOH estimation by fractional order models [9], electrochemical impedance spectroscopy [10] and estimation algorithms between battery internal resistance and available capacity based on equivalent circuit model extraction. These models are based on open-loop estimation and are limited by the quality of the test data. Therefore, some theories achieve better results by introducing closed-loop estimation such as the extended Kalman filter, particle filter and Lyapunov optimization. The extended Kalman filter is able to work under nonlinear system conditions by incorporating multiple estimation tools, resulting in better closed-loop estimation. These closed-loop estimation algorithms rely on accurate modeling to achieve reliable prediction, so an accurate model is important for these methods.

In recent years, data-driven algorithms have been applied to prediction tasks in SOH, and these methods are usually based on the extraction of available features and the flexible use of machine learning methods. Methods represented by Gaussian regression [11], genetic algorithms, support vector machine algorithms [12] and artificial neural networks [13] are used in SOH prediction tasks. Deep learning, as a special kind of machine learning, transforms complex systems into simple systems by learning nested combinations of simple linear structures to achieve the ability to fit and predict complex systems [14,15]. Unlike machine learning algorithms that rely on expert knowledge, deep learning is able to extract more combinations from simple features, and this feature has been widely used in the field of image recognition, where it has achieved good results. This feature has been widely used in image recognition. As a typical time-series model, recurrent neural networks (RNNs) have achieved good results in this class of models. The two most typical RNN variants are long short-term memory (LSTM) and gated recurrent network (GRU). The existing research on SOH prediction based on deep learning implementation, the data input method used is generally preprocessing the battery charging and discharging data sequences to some

extent and extracting typical values such as IC curve peaks to generate a set of feature vector sequences for input, so as to achieve SOH prediction. For complete charging data, this fixed feature extraction prediction achieves good results, but for incomplete charging data, the lack of feature vector dimension will have a great impact on the prediction effect of the model. Therefore, this method is not suitable for this study.

Based on the above SOH estimation results, the RUL is utilized as another battery health parameter to measure the remaining service time of the battery. By predicting when the cells will reach their end-of-life based on the historical trend of SOH or capacity, this information can be valuable for battery management and strategies for extending battery life. Qin et al. [16] proposed an RUL prediction method using particle filters and artificial neural networks. In Ref. [17], a new online RUL prediction using a smooth particle filter-based likelihood approximation method was introduced, and the results show that the proposed approach gives improved accuracy and improves the convergence rate. Additionally, an enhanced particle filter technology [18] and a physics-based model-informed smooth particle filter [19] were designed for RUL prediction.

### 1.3. Main Contributions

The calendar aging for lithium-ion batteries used in the UPS system is hard to estimate because of the slow decay rate of the battery, and it is difficult to find measurable decay characteristics. This paper proposes a particle-filtering-based algorithm for battery SOH and RUL estimation. Considering that the particle filtering algorithm itself has many adjustable parameters, this paper uses the grid method of hyperparameter search to find the parameters of the particle filtering algorithm applicable to UPS lithium-ion phosphate batteries and then performs SOH estimation and RUL prediction for the constructed single-cell battery calendar aging model, and finally, the single-cell SOH and RUL estimation algorithm is widely applied to the application scenarios of battery packs and groups. The experimental results indicated that the proposed method can achieve accurate SOH estimation and RUL prediction results. The main contributions of this work are as follows: (1) The calendar aging modeling for the batteries used in the UPS power supply system for an energy storage power station is presented, and the state space equations for the calendar aging model are established for state estimation. (2) The particle filtering algorithm is employed for the SOH estimation and RUL prediction. (3) The single-cell SOH and RUL estimation algorithm is expanded to the pack and group scales estimation.

### 1.4. Outline of the Article

The outline of the article is as follows: Section 2 gives the battery calendar aging model descriptions. Section 3 presents the SOH and RUL estimation algorithms, including the cell, pack and group scales. Section 4 presents the experimental results and discussions. Finally, the conclusions are given in Section 5.

## 2. Battery Calendar Aging Modeling

The calendar aging modeling is used to describe the battery capacity decay behavior of lithium iron phosphate batteries in an Uninterruptible Power Supply (UPS) under float charging conditions. The analysis of the calendar aging mechanism of the battery can lay the foundation for the subsequent SOH estimation of the battery. A reasonable calendar aging model is essential for accurate SOH estimation and RUL prediction, so a calendar aging model adapted to the working conditions of this paper needs to be established. The aging of lithium-ion batteries mainly consists of two aging modes: on the one hand, the loss of available recycled lithium due to the formation and growth of solid-phase electrolyte interface (SEI) film, and on the other hand, the loss of active material in the positive and negative electrodes, where the growth of SEI film is the main cause of calendar aging. According to Arrhenius' law, temperature plays an important role in the formation of SEI

films. Taking the capacity change of the battery as the main indicator of battery aging, the SOH of the battery is defined as:

$$SOH_k = C_k / C_0 \times 100\% \tag{1}$$

Here, an empirical model of calendar aging is given as follows:

$$y_k = A(SOC) \exp\left(-\frac{E_a}{RT}\right) k^z \tag{2}$$

where $k$ indicates the battery calendar aging time (days). $y_k = 1 - SOH_k$ denotes the battery capacity decay rate, $C_k$ is the current battery capacity and $C_0$ is the initial capacity. $A(SOC)$ is the pre-exponential term factor, which depends on the float charge SOC (float charge voltage). $E_a$ is the activation energy in $J \cdot mol^{-1}$. $T$ is the temperature in Kelvin ($K$). $R$ is the molar gas constant in $J/mol \cdot K$. $z$ is the density law coefficient. If the capacity decay of calendar aging is determined only by the SEI film growth, then $z = 0.5$. Considering other causes of calendar aging, the value here is to be determined.

In addition, based on the Supervisory Control and Data Acquisition (SCADA) data, it was found that the ambient temperature and float voltage of the UPS battery remained essentially constant, so the calendar aging model in this paper does not need to consider the effects of temperature and $SOC$ (float voltage). The above empirical model of calendar aging can be reduced to:

$$y_k = \alpha k^z \tag{3}$$

where $\alpha = A(SOC) \exp(-E_a / RT)$ is a constant in this work.

## 3. Methodology

### 3.1. Battery Cell SOH Estimation Algorithm

To implement battery cell SOH estimation, the calendar aging model should be established as a state space representation. Taking the logarithm of both ends of Equation (3), we can obtain:

$$\ln y_k = z \ln k + \ln \alpha \tag{4}$$

Its corresponding recursive form is as follows:

$$\ln y_k - \ln y_{k-1} = z \ln \frac{k}{k-1} \tag{5}$$

For computational purposes, a state permutation is performed as:

$$h_k = \ln(y_k) \tag{6}$$

Then, the calendar aging state space model is shown as follows:

$$\begin{bmatrix} h_k \\ z_k \end{bmatrix} = \begin{bmatrix} 1 & \ln \frac{k}{k-1} \\ 0 & 1 \end{bmatrix} \begin{bmatrix} h_{k-1} \\ z_{k-1} \end{bmatrix} + \begin{bmatrix} \omega_{h,k} \\ \omega_{z,k} \end{bmatrix} \tag{7}$$

$$y_k = \exp(h_k) + v_k \tag{8}$$

where Equations (7) and (8) are the state equation and observation equation, $\omega_{h,k}$ and $\omega_{z,k}$ are the process noise and $v_k$ is the measurement noise. The state quantity to be estimated is $x_k = [h_k, z_k]^T$.

Based on the calendar aging state space model, this scheme uses the particle filter algorithm for single-cell SOH estimation. The particle filter is a recursive nonlinear filter based on Bayesian principle using Monte Carlo method for systems containing non-Gaussian noise [20]. The main idea is to simulate the posterior probability distribution of time by multiple random particles with weights and thus estimate the state of the nonlinear system

from the observation sequence containing noise. The particle filtering algorithm specifically includes the following parts: particle initialization, importance sampling and resampling. The detailed procedure of the particle filtering algorithm is as follows:

Step 1: Particle initialization: for $k = 0$, the initial particles $Ns$ are generated by the prior probability density function: $\hat{x}_0^i, i = 1 \dots N_s$;

Step 2: Importance Sampling: for k = 1, 2, 3, . . .

(1) Update the particles: for any $i$, generate a new sample of states (particles) for the current $k$ moments according to the state Equation (7) and the previous moment particle population: $\hat{x}_k^i = A\hat{x}_{k-1}^i + Bu_k + \omega_k$. The battery capacity decay rate at moment $k$ can be obtained according to Equation (8): $\hat{y}_k^i = f\left[\hat{x}_k^i, u_k\right] + v_k$.

(2) Generating weights: based on the measured value of the battery capacity decay rate at moment $k$, the importance weights are generated according to the following equation: $q_k^i = \frac{1}{\sqrt{2\pi Q_v}} \exp\left(\frac{1}{2Q_v}\left(y_k - \hat{y}_k^i\right)\right)$ (where $Q_v$ is the measurement noise covariance).

(3) Weight normalization: $\overline{q}_k^i = q_k^i / \sum_{i=1}^{N_s} q_k^i$

Step 3: Resampling: Based on the normalized particle weights, a set of a posteriori particles can be generated using a polynomial resampling method: $\widetilde{x}_k^i, i = 1 \dots N_s$. The parameter estimation results can be expressed as a weighted sum of the updated particles: $x_k^i = \sum_{i=1}^{N_s} \overline{q}^i \widetilde{x}_k^i$.

### 3.2. Battery Pack and Group SOH Estimation Algorithm

In this section, battery pack SOH estimation algorithm will be discussed. In this work, Figure 1a shows the topological relationship between the battery pack and the battery cells. In this case, 2 cells are connected in parallel as 1 battery monomer, and 12 battery monomers are connected in series to become a battery pack. Since the series-connected cells have the same current but different voltages, the minimum SOH of the series-connected cell monomer determines the SOH of the pack, so the SOH of the pack can be defined as the minimum value of the SOH of the series-connected cell monomer:

$$SOH_k^{\text{pack}} = \min_{i=1-12}\left(SOH_k^i\right) \qquad (9)$$

where $SOH_k^{\text{pack}}$ is the SOH of the battery pack, $SOH_k^i$ is the SOH of the battery cells connected in series in the battery pack and $i$ is the serial number of the battery cells. Then, according to cell SOH estimation algorithm, the SOH of the battery cells can be calculated separately. Then, take the minimum value according to Equation (9), which is the SOH of the battery pack.

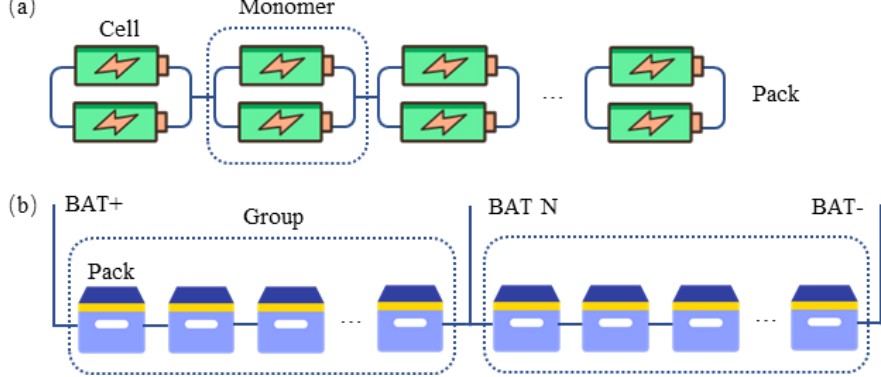

**Figure 1.** (**a**) Topological relationship between the battery pack and the battery cells. (**b**) Topology of the battery group and battery pack.

Figure 1b shows the topology of the battery group and battery pack in which seven battery packs are connected in series to become a battery group. The battery pack is divided into positive and positive battery groups. Without loss of generality, the SOH of the positive battery group (7 packs) is estimated here. The negative battery group is similar and will not be described.

Since a battery group is composed of 7 packs connected in series, the series packs have the same current but different voltages, the minimum SOH of the series packs determines the SOH of the group, so the SOH of the battery group can be defined as the minimum of the SOH of the series packs as follows:

$$SOH_k^{\text{group}} = \min_{j=1\sim7}\left(SOH_k^j\right) \tag{10}$$

where $SOH_k^{\text{group}}$ is the SOH of the battery group, $SOH_k^j$ is the SOH of the battery pack connected in series and $j$ is the serial number of the battery pack. Then, the SOH of the battery pack is calculated separately according to the previous section, and then the minimum value is taken according to Equation (9), which is the SOH of the battery group.

*3.3. Battery Cell RUL Estimation Algorithm*

The battery remaining useful life (RUL) represents the number of days remaining before the battery capacity decays to the EOL (end-of-life) condition. Here, the EOL of the battery cell is set to 50% of the nominal capacity of the cell, which means that the battery needs to be replaced when the capacity of the battery cell decays to 50% of its initial capacity.

Define the number of days used as $\mathbf{N}_{0:i} \triangleq \{n_0, n_1, \ldots, n_i\}$ and SOH sequence as $\mathbf{SOH}_{0:i} \triangleq \{soh_0, soh_1, \ldots, soh_i\}$, where $i$ is the particle number in the particle filtering algorithm above. Then, the EOL of the cell monomer is defined as:

$$N_{eol} = \inf\{n_k : y_k > EOL + v_k | y_i < EOL + v_i\} \tag{11}$$

The remaining days of use based on UPS usage conditions are defined as:

$$L_i = \inf\{l_i : \mathbf{SOH}_i > EOL | \mathbf{SOH}_{0:i}\} \tag{12}$$

where $n_k$ and $L_i$ are the number of current cycles and the number of days of remaining use, respectively. Since the prediction of RUL illustrates the long-term pattern of battery capacity variation, the overall trend is more important compared to the transient variation. Therefore, the SOH estimation cannot be directly used for the RUL prediction. This scheme uses a weighted average filtering algorithm to filter the SOH estimation results and then perform the RUL prediction. The specific procedure is as the following equations:

$$SOH_k^* = \sum_{i=k-win+1}^{k} \varpi \cdot soh_i \tag{13}$$

$$\varpi_i = \varpi_k + d(k-i) \tag{14}$$

$$\sum_i \varpi_i = 1 \tag{15}$$

where $SOH_k^*$ is the processed SOH value. $\varpi_i$, $d$ and $win$ are the weights, equal weight differences and sliding window sizes, respectively.

*3.4. Battery Pack and Group RUL Estimation Algorithm*

According to the topological relationship between the battery pack and the battery cells, the minimum value of the remaining days of use of the battery cells determines the

remaining days of use of the battery pack due to the series topology. Therefore, the RUL of the battery pack can be calculated by the following equation:

$$L_{\text{pack}} = \min_{i=1\sim12}\{L_i\} \tag{16}$$

where $i$ is the serial number of the battery cell in the pack, and $L_i$ is the number of days left in the battery cell.

According to the topological relationship between the battery group and the battery pack, the minimum value of the remaining days of use of the battery pack determines the remaining days of use of the battery group due to the series topology. Therefore, the RUL of the battery group can be calculated by the following equation:

$$L_{\text{group}} = \min_{j=1\sim7}\{L_j\} \tag{17}$$

where $j$ is the serial number of the battery pack in the battery pack, and $L_j$ is the number of days left in the battery pack.

The overall framework of the prediction algorithm is shown in Figure 2. This paper proposes a particle-filtering-based algorithm for battery SOH and RUL estimation. Considering that the particle filtering algorithm itself has many adjustable parameters, this paper uses the grid method of hyperparameter search to find the parameters of the particle filtering algorithm applicable to UPS lithium iron phosphate batteries and then performs SOH estimation and RUL prediction for the constructed single-cell battery calendar aging model, and finally, the single-cell SOH and RUL estimation algorithm is widely applied to the application scenarios of battery packs and groups to achieve accurate SOH estimation and RUL prediction.

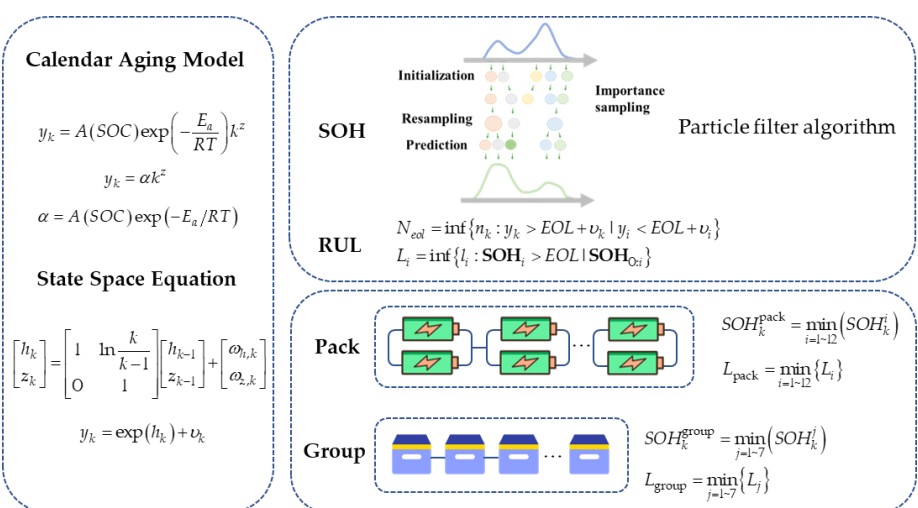

**Figure 2.** Framework of the prediction algorithm.

## 4. Results and Discussion

### 4.1. Experimental Testbench

The object of the experiment is the batteries used in the uninterruptible power supply system of the Shanghai rail transit energy storage power station with a capacity size of 100 Ah and a rated voltage of 3.6 V for a single one. For one battery group, there are seven battery packs constituted. For one independent battery pack, there are twelve single-cells constructed. Considering the complexity of the actual system and the fact that the designed SOH and RUL algorithms can be verified within a single battery pack in the lab, the subsequent experiments and results analysis are conducted for one single battery pack.

First of all, in this paper, the battery cells are tested by an eight-channel battery tester (NBT5V-200AC8-T) for charging and discharging for single-cell constant capacity testing.

All the experiments are conducted in a high and low temperature thermostat (SUYIDA GDW-100L) to control the experimental temperature. The purpose of the constant capacity testing is to establish the capacity discriminant table of the battery by determining the mapping rule between the terminal voltage of the battery and its SOC, which is used to perform the capacity consistency analysis of the battery, and to provide experimental reference values for the SOH and RUL estimation algorithm to close the loop to correct the estimated results of the algorithm state. The experimental steps of the constant-current experiment are as follows:

(1) Charge the battery cell at a constant current of 0.3 C (30 A) and reach the charging cutoff voltage of 3.6 V when the current voltage is recorded as the terminal voltage corresponding to SOC = 100%;

(2) Discharge the battery with a constant current of 0.5 C (50 A) and record the corresponding terminal voltage change to 2.6 V, and stop discharging when the discharge cut-off voltage is reached;

(3) Create a capacity discrimination table. According to the end voltage curve obtained in step (2), starting from the discharge moment. The corresponding voltage value is taken every 12 min as the end voltage corresponding to SOC = [90%, 80%, 70%, 60%. 50%, 40%, 30%, 20%, 10%, 0%].

Subsequently, capacity tests are performed. The purpose is to calculate the actual capacity of the battery through the discharging capacity and remaining capacity of the battery according to the capacity discriminant table obtained in the capacity fixing experiment (reaching more than 50% of the rated capacity of the battery is qualified) and to replace the battery unit with the discharging termination voltage lower than 2.6 V and then re-test it. The experimental steps of the verification capacity test are as follows:

(1) Start the experiment by charging the battery at a constant current of 0.3 C (30 A) and stop charging when the charging cut-off voltage of 3.6 V is reached to ensure that the battery is fully charged, i.e., SOC = 1.0;

(2) Record the float voltage of each individual cell (note: voltage at the end of the cell that is not offline);

(3) Start the constant current discharge, and record the starting voltage of each single battery (Note: offline battery terminal voltage; do not rest; immediately measure); the discharge current size is 0.5 C (50 A), and the battery discharge is 1 h (discharge 50%). Do not rest; immediately record the end voltage at the end of discharge to determine whether the single battery voltage is lower than 2.6 V. If it is lower than 2.6 V, stop the experiment and replace the substandard qualified battery. Otherwise, continue the experiment;

(4) Determine the capacity of the battery according to the capacity discrimination table in the constant capacity test.

Finally, calendar aging experiments are performed. The purpose of this experimental protocol is to investigate the effect of calendar aging on battery aging, to obtain aging data and to serve for battery SOH estimation and remaining life prediction of the battery. The experimental steps for calendar aging are shown below:

(1) Conduct constant current and voltage charging with a charge current of 0.3 C (30 A) and a charge cut-off voltage of 3.6 V, ensuring that the battery is fully charged, i.e., SOC = 1.0;

(2) Check the battery cell voltage. If it is not lower than 3.35 V, then continue to store at 25 °C and check the battery. If the cell voltage is lower than 3.35 V, the battery is charged at a 0.3 C constant current;

(3) Detect the battery cell voltage. If the battery cell voltage does not reach 3.6 V and the total battery pack voltage does not reach 580 V, then continue to charge. If the voltage of the battery cell reaches 3.6 V or the total voltage of the battery pack reaches 580 V, then stop charging and continue to store at 25 °C;

(4) Conduct battery capacity testing every half month. If the aging time reaches one month, conduct capacity verification test and judge whether the aging test termination condition is reached, i.e., if the battery capacity decays to 50% of the rated capacity. If the test termination condition is not reached, return to step (3); if the test termination condition is reached, end the experiment. If the aging time does not reach one month, continue to store the battery at 25 °C and return to step (3).

Based on the existing UPS solid battery, through the addition of intelligent battery orphan management system to achieve real-time collection and transmission of the battery pack internal unit voltage, current and other data, we wirelessly connected to the battery data cloud platform and through the network background account to view and record. Therefore, this paper adopts the battery pack test system (CT-4000-50V100ANTFA) to charge and discharge the battery pack for calendar aging and adopts the self-developed cloud-based BMS to collect the information of the battery pack as well as the battery singles, as showed in Figure 3.

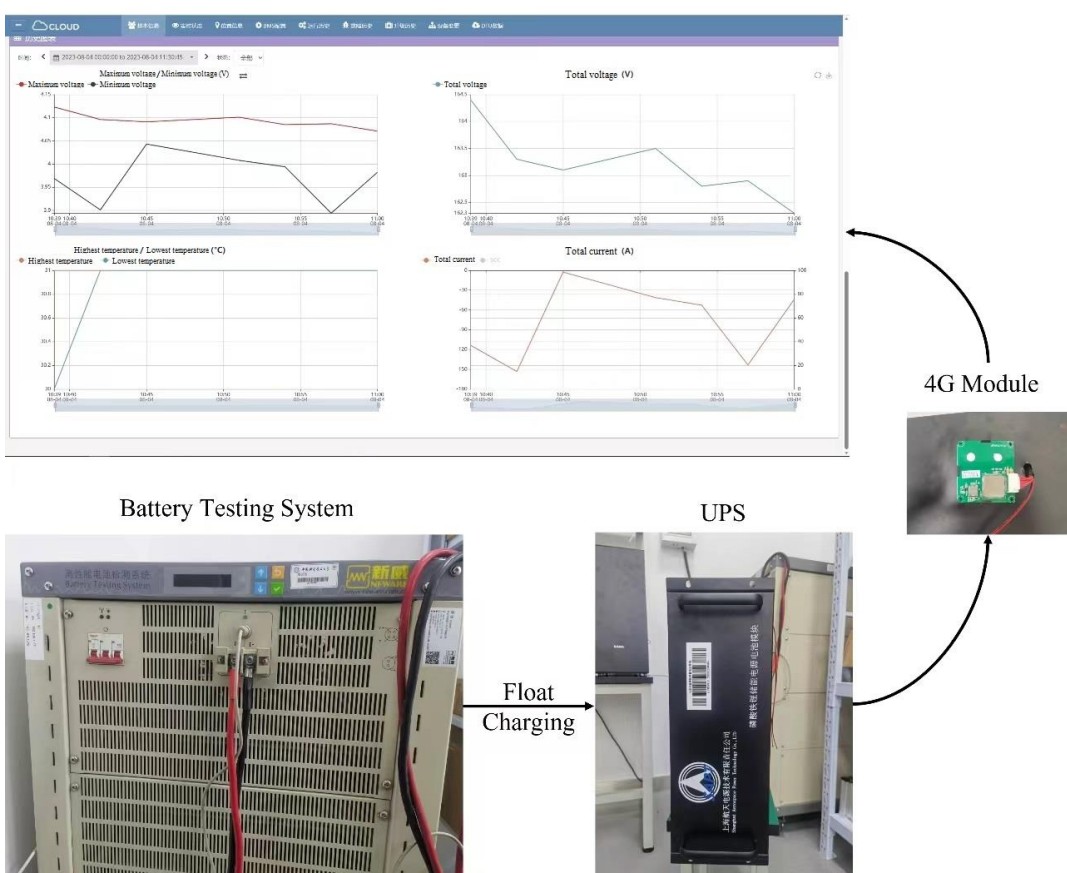

**Figure 3.** Scheme of Cloud-based BMS for UPS.

The actual capacity of the battery is calculated from the discharged capacity and the remaining capacity of the battery based on the capacity discrimination table obtained in the capacity determination experiment. In this experiment, capacity verification experiments were conducted every half month, and the time was maintained for about half a year. Thirteen capacity verification experiments were conducted, and a total of thirteen capacity points were obtained.

*4.2. Results and Analysis*

4.2.1. Battery SOH Estimation

Figure 4 gives the SOH variation curve of the single-cell under the calendar aging experiment. It can be seen that the SOH variation curves of all monomers are not monotonically decreasing, but there is a decaying trend that rises first, remains stable for a period of time and then starts to decrease. For batteries, as lithium-ion batteries are continuously activated in the early stage, also known as battery activation, the internal electrochemical properties of the anode and cathode materials and electrolyte. The internal electrochemical properties of the anode and cathode materials and electrolyte are activated, showing an increase in capacity, and SOH is maintained at 100%. As for Cell 7, its health status is obviously not as good as the rest of the cells, so it is assumed that the voltage sensor in the BMS is faulty or that Cell 7 is damaged.

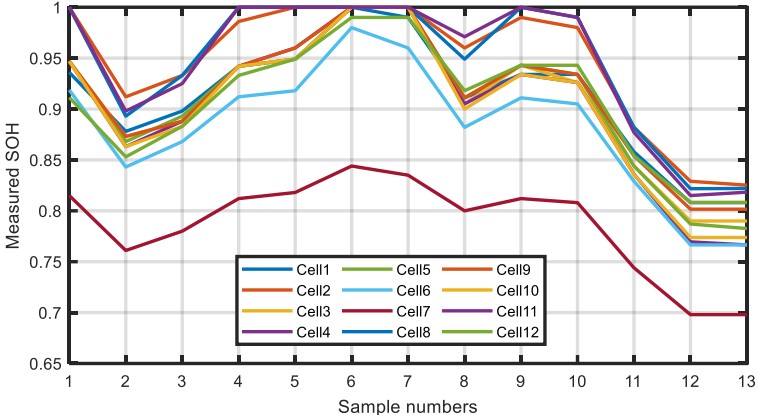

**Figure 4.** Battery cell SOH estimation results.

When setting the initial parameters of PF, $x_0 = [-2, 0.9]^T$, $\omega_{h,k}$ and $\omega_{z,k}$ are both set to $5 \times 10^{-2}$, and $v_k$ is set to $1 \times 10^{-5}$. The initial particle Ns is set to 200. As shown in Figure 5, the SOH estimation results are given for 12 single-cells in a pack. The horizontal coordinate is the cell SOH estimated using the PF algorithm, and the vertical coordinate is the experimentally measured value. The root mean square error (RMSE) and mean absolute percentage error (MAPE) of the prediction are given in Table 1.

**Table 1.** RMSE and MAPE of Battery cell SOH estimation.

| Cell | 1 | 2 | 3 | 4 | 5 | 6 | 7 | 8 | 9 | 10 | 11 | 12 |
|---|---|---|---|---|---|---|---|---|---|---|---|---|
| **RMSE (%)** | 1.87 | 7.26 | 2.97 | 3.12 | 3.58 | 1.8 | 0.15 | 6.37 | 3.1 | 3.15 | 9.07 | 2.06 |
| **MAPE (%)** | 1.18 | 5.37 | 1.85 | 2.17 | 2.79 | 0.97 | 0.16 | 5.42 | 1.96 | 2.07 | 6.85 | 1.58 |

The minimum value of SOH of the 12 individual cells under each measurement was subsequently taken as the SOH of the battery pack, as shown in Figure 6. The anomaly of Cell 7 caused a low estimate of the SOH of the battery pack.

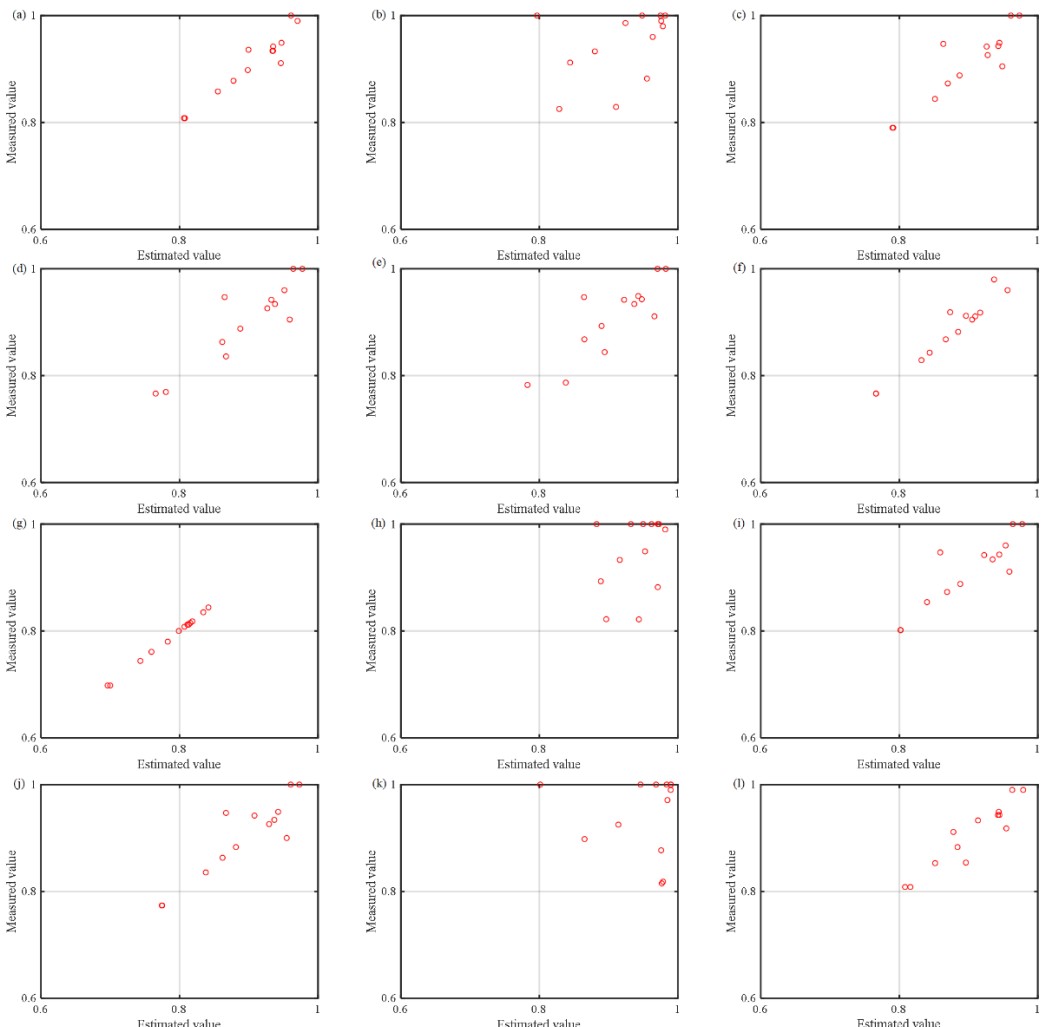

**Figure 5.** Battery cell SOH estimation results. (**a**–**l**) Battery SOH estimation for 12 single-cells in a pack.

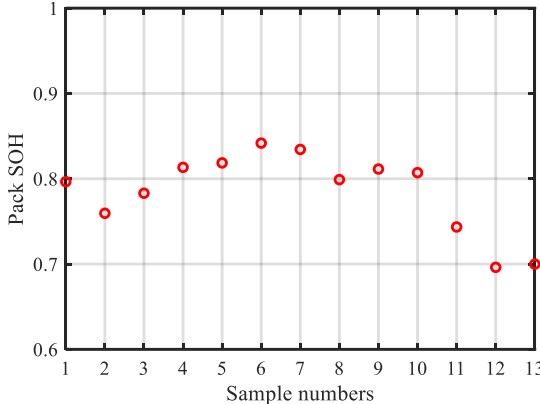

**Figure 6.** Battery pack SOH estimation result.

### 4.2.2. Battery RUL Estimation

The EOL of the battery cell is set to 50% of the nominal capacity of the cell, which means that the battery needs to be replaced when the capacity of the battery cell decays to

50% of its initial capacity. According to Equation (3), we can calculate $\alpha$ firstly based on the latest state variables and its output value.

$$\alpha = \frac{y}{k^z} \tag{18}$$

And then, we can set $y = 0.5$ and k will be calculated as

$$k = \left(\frac{0.5}{\alpha}\right)^{1/z} \tag{19}$$

Finally, as shown in Table 2, battery cell RUL estimation is given for each single-cell.

**Table 2.** Battery cell RUL estimation.

| Cell | 1 | 2 | 3 | 4 | 5 | 6 | 7 | 8 | 9 | 10 | 11 | 12 |
|------|------|------|------|------|------|------|------|------|------|------|------|------|
| RUL (Days) | 1231 | 1179 | 1950 | 1150 | 1759 | 1027 | 194 | 1099 | 1330 | 1841 | 1295 | 1425 |

The battery pack RUL estimation result is calculated through the minimum value of SOH of the 12 individual cells, which is 194 days.

## 5. Conclusions

Given the inevitable and continuous degradation of lithium-ion battery performance over time, the accurate estimation of the available capacity of lithium-ion batteries is critical to ensure their efficient and reliable operation. The calendar aging for lithium-ion batteries used in UPS system is hard to estimate because of the slow decay rate of the battery, and it is difficult to find measurable decay characteristics. This paper proposes a particle-filtering-based algorithm for battery SOH and RUL estimation. Considering that the particle filtering algorithm itself has many adjustable parameters, this paper uses the grid method of hyperparameter search to find the parameters of the particle filtering algorithm applicable to UPS lithium iron phosphate batteries and then performs SOH estimation and RUL prediction for the constructed single-cell battery calendar aging model, and finally, the single-cell SOH and RUL estimation algorithm is widely applied to the application scenarios of battery packs and groups. The experimental results indicate that the proposed method can achieve accurate SOH estimation and RUL prediction results.

**Author Contributions:** Conceptualization, W.X. and H.T.; methodology, W.X.; validation, H.T.; formal analysis, H.T.; investigation, H.T.; resources, W.X.; writing—original draft preparation, W.X. and H.T.; writing—review and editing, W.X. All authors have read and agreed to the published version of the manuscript.

**Funding:** This work was supported by the Shanghai Informatization Development Special Fund Project (No. 201901028).

**Data Availability Statement:** Not applicable.

**Conflicts of Interest:** The authors declare no conflict of interest. The authors are employees of Shanghai Electric Group Co. The paper reflects the views of the scientists and not the company.

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
