# Peer review of "Research on Calendar Aging for Lithium-Ion Batteries Used in Uninterruptible Power Supply System Based on Particle Filtering"

_wevj, doi:10.3390/wevj14080209_

Round 1

Reviewer 1 Report

This paper proposes a particle filtering-based algorithm for battery state-of-health (SOH) and remaining useful life (RUL) prediction. First, the calendar aging modelling for the batteries used in the uninterruptible power supply system for Shanghai rail transportation energy storage power station is presented. However, some slight modification suggestions are provided as follows:

1.      It is suggested that the author Improve the Abstract to increase the paper’s readability.

2.      Authors are suggested to give more description of the relevant background and to cite more relevant literature in recently. There are many approaches to estimate SOH and RUL; please give some deep and whole summarized. Some relevant examples:

-Remaining useful life prediction for lithium-ion batteries using particle filter and artificial neural network

 -Smooth particle filter‐based likelihood approximations for remaining useful life prediction of Lithium‐ion batterie

 - An enhanced particle filter technology for battery system state estimation and RUL prediction

-Physics-based model informed smooth particle filter for remaining useful life prediction of lithium-ion battery

3.      The authors should describe their contributions more clearly.

4.      The experiment lacks comparison with other typical methods, and the comparison should be wider than particle filter; it is hard to demonstrate that the proposed method has good efficiency.

5.      This paper uses RMSE as the metric for evaluating the performance of RUL prediction. However, there are a few other very popular and commonly used metrics, e.g., The absolute error, the relative error, Mean absolute percentage error (MAPE), Precision, Accuracy, cumulative relative accuracy (CRA) and adjusted R-Square. The authors also suggested using additional metrics for the performance evaluation.

English required improvement 

Reviewer 2 Report

Tan et.al reported the Research on Calendar Aging for Lithium-ion Batteries used in  Uninterruptible Power Supply System based on Particle Filtering. The manuscript is technically lacks in many aspects and reperesention of data, discussions is very eloborate. Hence, the manuscript can be consider for publication in world electric vechicle journal.

1.     Authors considered SEI growth considered main cause of calender aging. However, SEI can be considered only for anodes in lithium iron phosphate based batteries. How SEI can growth can affect the loss of active materials in the cathode side. Authors needs to provide relevant discussion on it. 

2.     Not only temperture plays major role in SEI formation. Electrolyte chemistry, electronic conductivity of anode active materials, etc., authors should compare extent of these reaction as well in SEI formation. 

3.     How these algorithm based methods differ from online measurement systems for SOH estimation. 

Moderate revision is required for better understanding.
